# The Effect of China’s Health Insurance on the Labor Supply of Middle-aged and Elderly Farmers

**DOI:** 10.3390/ijerph17186689

**Published:** 2020-09-14

**Authors:** Lingchen Liu, Renji Sun, Yan Gu, Kung Cheng Ho

**Affiliations:** 1School of Statistics, Shanxi University of Finance and Economics, Taiyuan 030006, China; liulingchen@fudan.edu.cn; 2Center for Population and Development Policy Studies, Fudan University, Shanghai 200433, China; 3Department of Global Health and Population, Harvard TH Chan School of Public Health, Boston, MA 02115, USA; 4The School of Business, East China University of Political Science and Law, Shanghai 201620, China; 2810@ecupl.edu.cn; 5Fanhai International School of Finance, Fudan University, Shanghai 200433, China; 13110680032@fudan.edu.cn; 6Pearl River Delta Collaborative Innovation Center of Scientific Finance and Industry, Institute of Regional Finance, Guangdong University of Finance & Economics, Guangzhou 510320, China

**Keywords:** health insurance, middle-aged and elderly farmers, labor supply

## Abstract

Social security primarily improves residents’ welfare and ensures labor market sustainability. This study presents a new view of the association between health insurance and labor supply by using data from the China Health and Retirement Longitudinal Study. The results reveal that the health insurance system has a remarkable effect on labor supply. The health insurance coverage tends to encourage middle-aged and elderly farmers to increase their farm labor participation rate and working time, especially for their household agricultural labor participation rate and working time. However, it also reduces the non-farm labor participation rate and working time. Different types of health insurance have diverse effects on labor supply. The new cooperative medical insurance has a stronger pull-back effect. It encourages the middle-aged and elderly farmers to leave the urban non-farm sector and transfer to rural areas to engage in their household agricultural work. The urban employee medical insurance encourages farmers to reduce self-employed labor supply and increase employed work. The supplemental health insurance tends to reduce the labor supply of farm employed and non-farm labor supply, but improve the farm labor supply. Furthermore, urban resident medical insurance and government medical insurance encourage farmers to quit directly from the labor market. In conclusion, the health insurance system is facilitating change in the labor market. Policy-makers should pay full attention to such impacts while improving the health insurance system’s design and operation in China.

## 1. Introduction

Based on the international experience, social security not only has the primary function of maintaining social stability and improving the welfare of residents but also has the vital function of ensuring the sustainable operation of the labor market. Since the mid-20th century, welfare systems that were established earlier, such as those in Germany, United Kingdom, the United States, and Japan, have begun to exert an increasingly obvious influence on the labor market while maintaining social stability and improving the national welfare system. These countries must cover ever-increasing medical expenses and pensions in the context of declining labor participation rates and labor supply time, especially of middle-aged and elderly adults. These two elements hamper the development of the economy, resulting in weak economic growth. These phenomena have caused scholars and relevant government departments to study the relationship between the social security system and the labor market.

In recent ten years, profound changes have taken place in the labor market. According to data released by the China National Bureau of Statistics, in 2019, the 16–59 working-age population was only 896 million, a decrease of 890,000 from the end of 2018. Since 2012, the absolute number of the working-age population has declined for nine consecutive years in China. The migrant worker tide turned to the migrant worker shortage, and the competition for talents became more and more fierce. With the rapid advancement of China’s industrialization and urbanization, rural laborers, especially young and middle-aged groups, in order to seek employment and increase income, abandon farming and move into cities for work, making rural middle-aged and elderly people the primary labor force in agricultural production. There is a severe shortage of China’s rural labor supply. At the same time, the reform of China’s social security system has accelerated. The New Cooperative Medical insurance was formally piloted in all provinces, autonomous regions, and municipalities in 2003. At present, comprehensive coverage has been achieved. Driven by rapid urbanization and labor mobility, some rural residents have participated in various types of medical insurance, such as urban employee medical insurance and urban resident medical insurance. In a short time, China has completed the comprehensive expansion of the medical security system from units to individuals, from urban areas to rural areas, and from employees to residents, and the level of treatment has been continuously improved. While medical security improves workers’ welfare, it also has an essential impact on their labor supply behavior. The middle-aged and elderly farmers are not only the main body of agricultural labor but also a group with relatively poor health and close to withdrawing from the labor market. They are more sensitive to the response of the medical security system. Like other countries that have established a comprehensive social security system earlier, China’s health insurance system might reduce the labor supply time of people or accelerate their withdrawal from the labor market, thereby accelerating the transformation of the labor market. This issue is not only related to the development and reform direction of China’s health insurance system itself, but also the sustainable development of China’s labor market and the long-term growth of China’s economy in the future.

This study is divided into five sections. The second section summarizes the relevant literature. The third section introduces the theoretical framework and econometric model, the fourth section presents the empirical results, and the fifth section provides conclusions, insights, and implications.

## 2. Literature Review

In general, health insurance reduces expected medical expenditures and preventive savings, thereby incentivizing workers to reduce labor supply. Studies have investigated the effect of health insurance on labor supply and retirement behavior earlier [1,2,3,4,5]. Boyle and Lahey [6] use the Current Population Survey (CPS) data study to reveal that participating in health insurance encourages older people to reduce labor supply time. Rogowski and Karoly [7] and Blau and Gilleskie [5] estimate the impact of U.S. social insurance on retirement decisions for older workers. The results indicate that social health insurance significantly reduces the participation rate of older workers and encourages older workers to withdraw from the labor market. Rogowski and Karoly [7] study the role of health insurance in retirement decisions for the older according to Health and Retirement Survey data. The results demonstrate that health care insurance encourages the older to reduce the labor participation rate, and the participation rate of employees participating in health insurance is 24% less than the participation rate without participation. Campolieti [8] uses the National Demographic Health Survey to study the effect of medical insurance on the labor participation of older men in Canada. The research reveals that medical security encourages older men in Canada to reduce labor participation. Chou and Staiger [9] draw similar conclusions on the study of non-working married women. Page [10] explores the impact of government medical insurance on the labor supply of kidney transplant recipients. The results indicate that increasing the coverage of health insurance reduce these groups’ labor participation rate. Bradley et al. [11] examine the effect of health insurance on married women workforce, arguing that married women participating in health insurance continue to work 8–11% less than non-participating. Liu [12] investigates the effect of the New Rural Cooperative Medical Scheme on farmers’ non-agricultural labor supply behavior in China and finds that New Rural Cooperative Medical Scheme significantly reduces non-agricultural labor force participation and employment.

Lowering the level of health insurance benefits may encourage workers to increase labor supply because of pressure from medical expenses. Evidence from the labor economics literature suggests that divorce may result in the loss of health insurance for a wife or the need to expand health insurance, resulting in divorced women increasing their labor supply [13]. Garthwaite et al. [14] examine how the loss of coverage in the most extensive public health insurance in the history of the United States affected labor supply. The results indicate that Tennessee residents who are losing public health insurance add at least 20 h of labor supply per week.

Health insurance may also stimulate an increase in labor supply. It plays a role in smoothing economic risks and improving workers’ health, which is conducive to improving the working capacity and labor compensation of workers. Health insurance will also encourage the insured to increase labor participation rate or labor supply time. Gruber and Hanratty [15] report that universal health insurance in Canada significantly motivates increased labor demand, probably because it improves labor mobility and labor health. Zheng [16] argues that health insurance plays a role in smoothing economic risks and ensuring health quality, thus improving the quality of labor of the insured. Moreover, when the level of health care is low, health insurance has a substitution effect on the insured who have higher expected labor income if they give up leisure, and the improvement of the medical level increases labor supply. Xu and Liu [17] use CHARLS data to estimate the impact of the new cooperative medical insurance on rural women’s labor supply. The study reveals that the proportion of hospitalization reimbursement at the new cooperative medical insurance and the annual inpatient capping line significantly encourages rural women to increase labor supply. Farooq and Kugler [18] use Current Population Survey Data and reveal that Medicaid increases occupational and industrial mobility. Kandilov and Kandilov [19] do not find evidence that the ACA Medicaid expansions led to a decline in farmworkers’ labor supply, but note a small increase in farmworkers’ labor hours. Several studies verify that health insurance plays a significant role in improving folks’ health status [20,21,22]. Many studies highlight that improved health status not only increases farmers’ labor supply but also actively reduces rural poverty or increases farmers’ income [23,24,25].

The effect of health insurance on labor supply varies among different groups. Because of individual characteristics, the effect of medical insurance on the labor supply of different groups also differs. Gruber and Madrian [26] find that health insurance has no effect on low-income mothers’ labor supply but plays a major role in labor supply decisions for middle-income earners. Murasko [27] examines the effect of spouse insurance on the labor supply of married women on the basis of panel data from the 1996–2004 Medical Expenses Panel Survey. The results indicate that the husbands’ health insurance had a strong and negative impact on wives’ working hours. Cebi and Wang [28] explore the relationship between a husband’s health insurance and a wife’s labor supply, and the results reveal that the spouse’s health insurance coverage reduced the wives’ labor supply. Tomohara and Lee [29] report that the National Child Health Insurance Plan encourages married women to reduce their working hours. Boyle and Lahey [6] use data from the U.S. Veterans Health Insurance in the mid-1990s to compare the labor market behavior of elderly male veterans and their non-retired wives before and after veterans’ health benefits according to the difference in differences method. The study demonstrates that health insurance significantly reduces the labor supply of men, whereas the labor supply of spouses increases. Aslim [30] investigates the effect of the Affordable Care Act’s Medicaid expansion on the retirement decision of low-educated adults aged 55–64 years and finds that the expansion results in women retiring early. By contrast, no significant change is noted in the retirement behavior of men. Beuermann and Pecha [31] estimate the effects of Jamaica’s elimination of user fees in public health facilities on the health and labor supply of working-age individuals. The analysis reveals no effects among individuals younger than 40 years. However, for individuals within the 40–64 year range, the policy reduces the number of lost days due to illness by 44.3%, and overall benefits are relatively more substantial for women.

Some studies have shown that health insurance has an insignificant effect on labor supply. Bailey and Chorniy [32] use the 2008–2013 Current Population Survey to estimate the magnitude of the job lock effect for young adults and find that the expansion of dependent coverage did not increase job mobility. Frisvold and Jung [33] use the 2011–2015 March Current Population Survey Supplements to compare the changes in insurance coverage and labor market outcomes over time of adults in states that expanded Medicaid and in states that did not. The results reveal that the expansion of insurance coverage does not impact labor force participation and labor supply time. Using a group of uninsured low-income adults in Oregon that is selected by lottery for the chance to apply for Medicaid and administrative data. Baicker et al. [34] find no significant effect of Medicaid on employment. Therefore, the overall effect of the labor supply of medical insurance still needs more in-depth and extensive empirical research.

Through the first paragraph of the literature, developed countries that established social security system earlier are bearing the burden of continuously increasing medical fund, and at the same time are faced with the continuous decline of labor participation rate and long-term sluggish economic growth, which leads to the discussion on the relationship between the health insurance system and the labor market. Currently, China’s health insurance system is still in the stage of frame design and system exploration, especially the newly established rural health insurance system. Policies and research departments mainly focus on the construction and implementation of the social security system itself, but the impact of the labor supply has not yet attracted enough attention. However, the relevant research has paid little attention to the impact of health insurance on farmers’ labor supply behavior. Therefore, the study of this weak field in China is of great theoretical and practical significance. The review of literature therefore suggests that the impact of health insurance on labor supply is mixed. These conflicting conclusions could be attributed to the substitution effect and income effect of medical treatment on labor supply. Because China’s social security system is still in the initial stage of reform and development, the research on the impact of China’s health insurance system on labor supply is relatively scarce, and the research on the labor supply effect of different types of health insurance projects is rare. This study presents a new view of the association between health insurance and labor supply by using data from the China Health and Retirement Longitudinal Study (CHARLS) in 2013. In contrast to the earlier literature, which focused on the labor supply effect of health insurance, we investigated the effect of the health insurance, whether treated as a whole or by various types of insurance on labor supply, such as urban employee medical insurance, urban resident medical insurance, and supplement health insurance. Second, the literature on the impact of health insurance on farmers’ supply of part-time labor is even rarer. In fact, due to the Chinese dual economic structure, it is normal for farmers to do part-time labor. We disaggregate the labor supply into four different types: farm employed, household agricultural work, employed work, and self-employed work to examine the impact of health insurance on the labor market. Third, we develop a stochastic two-period model to illustrate the potential effects of health insurance on labor supply.

## 3. Theoretical Framework and Econometric Model

### 3.1. Theoretical Framework

This section considers a two-stage model based on Bai and Wu [35] and Wang and Peng [36] utility models to explain the impact of health insurance on labor supply. Equation (2) expresses the constraints of the first period. Equation (3) is the second-period constraint. Use of health services improves health and working capacity and reduces sick days and potentially increases hourly wage. *Y*_1_ is the labor income, *C*_1_ is consumption, *S* is savings, *I* = {0,1} is participation in health insurance decision-making, *e_j_* (*j* = 1, 2, 3, 4, 5) is health insurance premium rate, where (*j* = 1, 2, 3, 4, 5) denotes a new cooperative, urban employee medical, urban resident medical, government medical and supplement health insurance respectively; *i* is the interest rate. Equation (3) is the second constraint. *Y*_2_ is income, *C*_2_ is consumption. *M* is medical expenses and is a random variable, *α*(*M*)*_j_* is the health insurance reimbursement amount, and (1 + *i*)*S* is savings. Equation (4) is the time budget constraint where the time is allocated for farm-employment, household agricultural work, employment, self-employment time and leisure time. With the increase in agricultural labor productivity, labor will gradually shift towards sectors with higher incomes. This shift begins in the field of agricultural diversification, family manufacturing industry, and firms established in local townships. Then, labor will shift to urban areas, some farmers will relocate to cities, and some they may even change their farmer status to urban workers or citizens. But more farmers will not cease farming altogether. They became migrant workers and switch back and forth between farming, working, and merchandising flexibly. Many migrant farmers are considered to be employed in the informal sector. Within this group, households with agricultural work as the main source of income are Part-time agricultural households, farmers with the main source of income other than agricultural work are part-time non-agricultural households. Farmers with agricultural work as a pure source of income are pure farming households, i.e., Farmers that are completely engaged in non-agricultural production are non-agricultural production households. Farmers completely dissociate from the field of production and operation are called non-operating households, i.e., In our theoretical model, it is assumed that the farmers will consider their endowments and choose the right type of part-time jobs or statuses and make choices between times to maximize self-interest. Equation (5) is the income source of each period, including the employed farm income $w_{{}_{t}}^{f_{1}}LS_{{}_{t}}^{f_{1}}$, the income from household agricultural work $p_{t}^{Y_{f_{2}}}Y_{{}_{t}}^{f_{2}}-p_{t}^{X_{f_{2}}}X_{{}_{t}}^{f_{2}}$, the employed income $w_{t}^{n_{1}}LS_{t}^{n_{1}}$ and the self-employed income $p_{t}^{Y_{n_{2}}}Y_{t}^{n_{2}}-p_{t}^{X_{n_{{}_{2}}}}X_{t}^{n_{2}}$, and other net transfer income $Q_{t}$. $w^{n_{1}}$, $w^{f_{1}}$ are wages, and $p^{Y_{f_{2}}}$
$p^{X_{f_{2}}}$ are the prices of output $Y^{f_{2}}$ and input $X^{f_{2}}$ of household agricultural work. $p^{Y_{n_{2}}}$, $p^{X_{n_{2}}}$ are the price of labor output $Y_{}^{n_{2}}$ and input $X_{}^{n_{2}}$ in the self-employed activities. Equation (6) represents the household agricultural production function. $Z^{f_{2}}$ represents other production factors, such as land, human capital, etc. Equation (7) is the self-employed production function and $Z^{n_{2}}$ represents other factors of production, such as factories, human capital, etc. The life-long utility of workers $U\left(C_{1},L_{1}\right)+\beta EU\left(C_{2},L_{2}\right)$ is that consumption and health insurance are regarded as regular products, and lifelong utility function is established:(1)$$\max\left[U\left(C_{1},L_{1}\right)+\beta EU\left(C_{2},L_{2}\right)\right]$$
(2)$$s.t.C_{1}+S+e_{j}I=Y_{1}$$
(3)$$C_{2}+M-I\alpha {\left(M\right)}_{j}=Y_{2}+\left(1+i\right)S$$
(4)$$T_{t}=LS_{{}_{t}}^{f_{1}}+LS_{{}_{t}}^{f_{2}}+LS_{t}^{n_{1}}+LS_{t}^{n_{2}}+L_{t}$$
(5)$$Y_{t}=w_{t}^{f_{1}}LS_{t}^{f_{1}}+\left(p_{t}^{Y_{f_{2}}}Y_{t}^{f_{2}}-p_{t}^{X_{f_{2}}}X_{t}^{f_{2}}\right)+w_{t}^{n_{1}}LS_{t}^{n_{1}}+\left(p_{t}^{Y_{n_{2}}}Y_{t}^{n_{2}}-p_{t}^{X_{n_{{}_{2}}}}X_{t}^{n_{2}}\right)+Q_{t}$$
(6)$$Y_{t}^{f_{2}}=f\left(LS_{t}^{f_{2}},X_{t}^{f_{2}},Z_{t}^{f_{2}}\right)$$
(7)$$Y_{t}^{n_{2}}=g\left(LS_{t}^{n_{2}},X_{t}^{n_{2}},Z_{t}^{n_{2}}\right)$$
(8)$$I=\left\{0,1\right\},C_{1},C_{2}>0,t=1,2$$

We will no longer optimize the theoretical model, focusing on explaining the mechanism of health insurance on labor decision-making from the theoretical framework. Assuming that the optimal labor supply quantity is *LS*(0) when workers do not participate in health insurance (*I* = 0), and the optimal labor supply quantity is *LS*(1) when they join health insurance (*I* = 1).

Assuming the difference is *D*(*LS*) = *LS*(1) − *LS*(0), it can be decomposed into:(9)$$D\left(LS\right)=\underbrace{h\left(e_{j}-E\left(r{\left(M\right)}_{j}\right)\right)}+\underbrace{LS\left(1\right)-LS\left(0\right)-h\left(e-E\left(r{\left(M\right)}_{j}\right)\right)}\;\;\mathrm{income}\;\mathrm{effect}\;\mathrm{substitution}\;\mathrm{effect}\;\mathrm{of}\;\mathrm{health}\;\;\mathrm{isurance}\;\mathrm{on}\;\mathrm{labor}\;\mathrm{supply}$$

This paper mainly examines the impact of health insurance on labor supply and we focus on explaining the mechanism of the effect of health insurance on labor decision-making from a theoretical framework viewpoint, which includes two major effects. The first is the income effect. Health insurance reduces expected medical expenditures and preventive savings, namely, relaxing income budget constraints, encouraging farms to reduce labor participation or labor supply time, and devote more time to leisure. So, the income effect will lower the labor supply. The second effect is the substitution effect. The health insurance can also increase the health level and human capital of the insured by expanding the medical expenditures of the insured, which will improve the labor remuneration of the insured and encourage them to increase labor participation or labor supply time. Whether the health insurance policy increases or decreases the labor supply mainly depends on comparing the two effects. If the substitution effect dominates, the overall effect on the labor participation rate will be positive, leading to increased labor supply. If the income effect dominates, the overall effect will be negative, and the labor participation rate and the labor supply level will decrease. If the two are equal, the combined effect is not apparent. Among them, *h*(×) is the income effect of medical security, which is an increasing function of the difference between the medical reimbursement amount *r*(*M*)*_j_* and the premium *e_j_*.

Apart from income and substitution effects, different medical insurance schemes *j* also exhibit different characteristics. First of all, different insurance schemes show discrepancies in fundraising methods. In urban employee medical insurance, the employer’s contribution is controlled at around 6% of salary and the employee’s contribution is 2% of salary. All contributions by employees are included in personal accounts. Contributions by the employers are divided into two parts. One part is used to establish an overall fund, and the other part is included in the personal accounts. The reimbursement rate is dependent on medication, medical-grade, the usual rate is around 70%. Urban resident medical insurance is funded mainly by individual (family) payments with some government subsidies. The level of protection and coverage is proportional to the amount of premium paid. New cooperative medical insurance is funded by individuals, the collective, and the government. The payment levels are stratified, the more farmers pay, the more protection and coverage they get. The government medical insurance is funded by the government and employers. Individuals are not required to contribute in this scheme. The government medical insurance scheme pays high reimbursement rates in claims. Funding for supplemental health insurance is raised by individuals, the collective, and the government. It covers medical needs beyond the scope of basic medical insurances. Although the financing method of different schemes varies with different levels of subsidies, the initial investment in the first period is small. This is helpful in reducing expected risks in later stages, reducing poverty caused by illness and smoothing consumption. From Equations (2) and (3), due to the initial investment in insurance, there will be more income left for consumption in period two. In addition, whether or not a farmer will work in the second period is uncertain. It is related to the severity of the disease he or she may catch. Thus, the potential income and health are related, and the potential health is related to the frequency and severity of the disease.

From Equation (2), *I* = 0 indicates no participation in insurance schemes, farmers are not required to pay the premium. Naturally, he or she will not receive claims or reimbursement in Equation (3). Non-insured farmers are required to save for themselves and their children. Once non-insured farmers caught significant illnesses, they are required to pay a substantial amount for a medical treatment which will in turn cause poverty, reduce labor supply, and ultimately withdraw from the labor market. From the literature, the proportion of poverty caused by illness in non-insured farmers is much greater than insured farmers [12]. Empirically testing whether *D*(*LS*) of Equation (9) is positive or negative is precisely the goal of the empirical section of this paper.

### 3.2. Econometric Model

The explanatory variables of the labor participation model are TLP, LPf, LPf1, LPf2, LPn, LPn1, LPn2, indicating total labor participation, farm labor participation, farm employed labor participation, household agricultural labor participation, non-farm labor participation, employed labor participation and self-employed participation. The explanatory variable is a binary selection variable of 0 or 1. We use the binary selection model Probit. To solve the possible heteroscedasticity problem, we use a robust standard deviation. The explained variables of labor supply model are TLS, LSf, LSf1, LSf2, LSn, LSn1, and LSn2, which indicate the total labor supply time, farm labor supply time, farm employed labor supply time, household agricultural labor supply time, non-farm labor supply time, employed labor supply time and self-employed labor supply time. It is a variable higher than or equal to 0. Considering that the labor supply time is a left-tailed limited explanatory variable, we choose the Tobit model as the estimation method.

For testing, the labor participation rate measurement model and labor supply time measurement model are constructed as follows:(10)$$LP_{{\left(T,f,f_{1},f_{2},n,n_{1},n_{2}\right)}_{i}}=\alpha_{0}+\alpha_{1}\cdot insurance_{i}+\alpha_{2}\cdot M_{i}^{s}+\alpha_{3}\cdot M_{{}^{i}}^{h}+\varepsilon_{i}$$
(11)$$LS_{{\left(T,f,f_{1},f_{2},n,n_{1},n_{2}\right)}_{i}}=\alpha_{0}+\alpha_{1}\cdot insurance_{i}+\alpha_{2}\cdot M_{i}^{s}+\alpha_{3}\cdot M_{{}^{i}}^{h}+\varepsilon_{i}$$

In the econometric model above, *i* is the *i*-th sample, *LP* and *LS* indicate the labor participation and labor supply, respectively. The subscripts *T*, *f*, *f*_1_, *f*_2_, *n*, *n*_1_, and *n*_2_ indicate the total labor participation, respectively, labor participation of farm, farm employed labor participation, household agricultural labor participation, non-farm labor participation, employed labor participation, and self-employed participation decision. LP is a binary variable with 0 and 1. Similar to many works of literature, Probit is used to estimate this model [37]. Comparing with traditional methods, PSM matches samples using a propensity score that eliminated selection bias. Therefore, PSM can be used as robustness checks. At the same time, IVprobit is added to account for endogeneity associated with cross-sectional data. L.S. is a continuous variable that is greater than or equal to 0, so the Tobit model is selected for estimation. IVtobit and PSM methods are used to overcome endogeneity and selection bias. $insurance_{i}$ indicates whether to participate in health insurance, $M_{i}^{s}$ express personal characteristic variables, $M_{i}^{h}$ represent family characteristic variables.

### 3.3. Variable Definition and Descriptive Statistics

According to the analysis of relevant literature and models (10) and (11), the main factors affecting the labor participation rate and labor supply time of the middle-aged and elderly people are the health insurance system, personal characteristics, family characteristics, and the region. This paper uses CHARLS data. Based on the national survey of CHARLS has three waves, respectively in 2011, 2013 and 2015, because the data in 2013 provide more abundant types of farmers’ part-time jobs than in 2015, this paper adopts the CHARLS of 2013. The CHARLS covered 150 counties, 450 villages, and 17,000 individuals in around 10,000 households, which is nationally representative. The national tracking data was collected across 28 provinces, and 18,605 final samples were obtained. The participation rate and labor supply of young people who have social security and those who do not are similar. However, middle-aged and elderly individuals are not only the main part of agricultural labor, but also the group tends to have poorer health, and they may also be the group that is close to exiting the labor market. Thus, they are more sensitive to social security policies [37]. Therefore, middle-aged and elderly individuals are the target of our interest. This paper focuses on farmers from 45 to 65 years old. After excluding missing values and outliers, 9875 effective rural samples were obtained. Specific variable definitions and descriptive statistics are shown in Table 1. The selection and processing methods of each variable are as follows:

Labor supply.

This article’s primary objects are the labor supply behaviors of middle-aged and elderly farmers, including labor participation and labor supply time. Labor participation is a variable of 0 or 1, and the unit of labor supply time is hour/year, from 2013 The work, retirement, and pension modules in the CHARLS in the national follow-up survey of the year are calculated according to the four aspects of farm employed, household agricultural work, employed work, and self-employed work. This article assumes that the investigation object is engaged in part-time work, which mainly includes four forms of part-time work, namely, farm employed, household agricultural work, employed work, and self-employed work. These four types of labor supply are activities to make money, and unpaid labor, such as volunteers, is not included. Labor participation is divided into total labor participation (TLP), farm labor participation (LPf), including farm employed (LPf1) and household agricultural work participation (LPf2)), and non-farm labor participation (LPn), including employed work (LPn1) and self-employed work (LPn2)). Labor supply time is divided into total labor supply time (TLS), farm labor supply time (LSf) including farm employed labor supply time (LSf1) and household agricultural work labor supply time (LSf2), and non-farm labor supply (LSn) includes employed labor supply (LSn1) and self-employed (LS2n).

Health insurance variables.

Due to data limitations, the health insurance variables only study the impact of the health insurance system on labor supply from the perspective of health insurance coverage. Health insurance coverage is subdivided into new cooperative medical insurance coverage, urban employee medical insurance coverage, urban resident medical insurance coverage, government medical insurance coverage, and supplement health insurance coverage. The health insurance system variables are judged from the medical care and insurance module in the 2013 National Tracking Survey CHARLS. The medical insurance coverage is a composite of the above various health insurance items.

Personal characteristic variables.

These are control variables and they mainly include sex, age, the square of age, education level, marital status, party members, and health status. Among them, sex, age, age square, education level, marital status, and whether they are party members are calculated or judged from the basic information modules in the 2013 National Tracking Survey CHARLS. Health status uses whether there is a daily activity disorder (ADL/IADL) as a proxy variable. This article does not use self-rated health, a more commonly used health measurement index because self-rated health is too subjective and prone to endogenous problems. This indicator is calculated according to the ADL/IADL table from the health status and function modules in the 2013 National Tracking Survey CHARLS.

Family characteristic variables, as control variables, they mainly include family size, family dependency ratio, total income, total income, farm income, farm employed income, household agricultural income, non-farm income, employed income, and self-employed income, area of cultivated land, the value of own house, the value of durable consumer goods, the total amount of bank deposits and cash, the financial support outside the family and the provision of family financial assistance. The family dependency ratio refers to the ratio of the number of children under 16 years of age and the number of older people over 60 years of age to the working-age population among the surveyed family members, reflecting the population coefficient of the average support of the respondents. The family size and family dependency ratio are calculated from the family module in the 2013 National Tracking Survey CHARLS. Total income, agricultural income, agricultural work, self-owned agricultural income, non-agricultural income, employment income, and self-employed income are calculated from the work, retirement, and pension modules in the 2013 CHARLS data. The cultivated land area, the value of self-owned houses, the value of consumer durables, the total amount of bank deposits and cash are calculated from the income, expenditure, and asset modules in the 2013 CHARLS data. Obtaining financial support outside the family and providing family financial assistance is calculated from the family interaction and financial assistance part of the family module of the CHARLS data in 2013.

Regional variables.

As a control variable, this article introduces three dummy variables in the east, middle, and west parts. Due to the vast differences in the development of various regions, especially economic development, there are vast differences in employment opportunities and income.

The labor participation and labor supply time impact of middle-aged and elderly people are also different. Regional variable indicators are judged from the first two in the survey subjects’ I.D. data in the 2013 National Tracking Survey CHARLS.

## 4. Results

### 4.1. Statistical Analysis of Health Insurance and Labor Supply

The total labor participation rate of middle-aged and elderly farmers participating in health insurance is higher than the total labor participation rate without participating. It can be seen from Table 2 that the total labor participation rate of participating in health insurance is 82.19%, while the total labor participation rate without health insurance is 77.27%. Mainly, the household agricultural labor participation rate of participating in health insurance is 68.82%, which is approximately 13% higher than non-health insurance participation. On the contrary, the non-farm labor participation rate of participating in health insurance is relatively lower. Further observing the different types of labor participation rates, we can see that the household agricultural labor participation rate in health insurance is relatively higher, while the farm employed labor participation rate, employed labor participation, and the self-employed participation rate is relatively lower.

The total labor supply time of middle-aged and elderly farmers participating in health insurance is also higher than that of non-participating in health insurance. It can be seen from Table 3 that the total labor supply time is 1318.5 h per year, which is higher than that of the non-participating in health insurance. The farm labor participation rate is particularly apparent. The farm labor supply time of participating in health insurance is 772.2 h per year, much higher than the farm labor supply time of not participating in health insurance. However, the non-farm labor supply time shows opposite statistical characteristics, and the supply time of non-farm labor participating in health insurance is relatively lower. To further observe the different types of labor supply time, the household agricultural labor supply time for participating in health insurance is relatively higher, while farm employed labor supply time, employed labor supply time, and self-employed labor supply time are relatively lower. Whether these statistical characteristics have significant causality needs to be further tested and analyzed by standardized econometric models.

There are significant differences in the labor participation rates among the middle and old aged farmers who participate in different health insurance projects. Table 4 shows that the total labor participation rate of middle-aged and elderly farmers participating in urban resident medical insurance is the lowest (72.73%), and that of urban employee medical insurance is the highest at 82.94%. The farm labor participation rate in urban employee medical insurance is the lowest, while the farm labor participation rate in supplement health insurance is the highest, 70.07%, followed by the new cooperative medical insurance. The farm employed participation rate in government medical insurance is 0, while the farm employed participation rate in supplement health insurance is the highest. The household agricultural work participation rate in the urban employee medical insurance is the lowest, while in new cooperative medical insurance is the highest. The non-farm labor participation rate in the supplement health insurance is the lowest, while that of urban employee medical insurance is the highest, at 65.88%. The employed participation rate in Supplement health insurance is the lowest, while that of middle-aged and elderly people who participate in urban employee medical insurance is the highest. The self-employed participation rate in the urban employee medical insurance is the lowest, while health supplement insurance in serious illness is the highest.

Similarly, there are also significant differences in the labor supply time of the middle-aged and elderly farmers participating in different health insurance projects, as shown in Table 5. These statistical differences reflect the differences in labor participation rates and labor supply times for different health insurance types. Therefore, it is necessary to verify further the labor supply effect of different health insurance systems.

### 4.2. Estimation of Labor Participation Model for Health Insurance Coverage

The health insurance coverage on the labor participation behavior of middle-aged and elderly farmers is not apparent. A possible reason may be that while medical insurance stimulates agricultural labor, it also reduces non-agricultural labor. Thus, the effect on the overall labor supply is not significant. However, health insurance exerts a structural change in the labor supply. Health insurance incentivizes farmers to reduce self-employed activities and leave the urban region, and ultimately flow back to rural areas for agricultural labor. The reason for such change may be that the new cooperative medical insurance only permits local claims and prohibits cross-regional claims. This rule means that farmers who wish to enjoy the benefits of insurance must abandon self-employed economic and non-agricultural labor. It is beneficial for rural development and safeguarding crop production.

From the estimated results (Table 6), in the household agricultural labor participation (LPf2), the health insurance participation variables pass the test at the significance level of 1%. The marginal effect estimated by the Probit Model is positive, indicating that participating in health insurance tends to increase the farm labor participation rate. From the perspective of the Probit marginal effect, participating in health insurance increases farm labor participation rate by 11.96%, household agricultural labor participation rate by 12.80%.

The coverage variables of health insurance in the non-farm labor participation model (LPn) and self-employed labor participation model (LPn2) all appear to be statistically significant, and the estimated marginal effects are negative, which indicate that under the control of other factors, participating in health insurance tend to reduce the non-farm and self-employed labor participation rate. From the marginal effect estimated by Probit, participating in health insurance reduces the participation rate of non-farm labor by 6.17%, the self-employed labor participation rate by 5.33%.

In the farm employed labor participation model (LPf1) and the employed labor participation rate (LPn1), the health insurance coverage variables do not appear to be statistically significant, indicating that participation in health insurance is not associated with the farm employed and employed labor participation decision-making behavior of middle-aged and elderly farmers.

### 4.3. Robustness Check

The endogeneity problem may arise with medical insurance coverage, labor participation decision, and labor supply. We address this issue by introducing whether or not the village where the family located carried out the new cooperative medical insurance as an instrument. This paper adopts IVprobit and IVtobit method for instrumental variable analysis. IVprobit is used to estimate the labor participation model, and IVtobit is used to estimate the labor supply model. The use of whether or not the village where the family located carried out the new cooperative medical insurance as an instrument variable is well documented in other research focusing on the new cooperative medical insurances [38]. It is worth mentioning that the 2013 CHARLS survey did not present community data, so we used individuals’ responses regarding the new cooperative medical insurance scheme to confirm whether his or her village carried out this insurance scheme. The Wald test result of this core explanatory variable to show exogenous hypothesis of insurance coverage is presented in Table 7. As shown, the *p*-value is less than 0.05, which rejects the null hypothesis and indicating an endogeneity problem. However, the direction obtained by IVprobit and Probit estimation method is the same.

This analysis shows that although the endogeneity problem does exist, it does not hinder the validity of the final results. Furthermore, considering the complexity and discrepancies across insurance schemes, this paper abandoned the instrument method and chose Probit and Tobit estimation to ensure the stability and comparability of the results between before and after.

Selection bias and other accidental factors may arise if we observe the effect of medical insurance on labor supply directly. We utilize the propensity score matching (PSM) method to account for such biases to ensure the correct estimation of medical insurance on farmers’ decision behavior on labor supply in Table 8. According to the logic of PSM, the probability model of the farmers’ decision on participating in medical insurance is estimated via the Logit method with the same control variables as the labor participation model. The results show that farmers’ education, marriage status, health, income, external support, household, and region factors show a significant influence on farmers’ decision-making process in whether or not to participate in medical insurances. Using this result, the propensity scores of the treatment and control group are matched, and average treatment on the treated (ATT) is computed.

Under matching pairs of the treatment and control group (Table 9), the difference in labor participation rate between farmers with insurance and farmers without insurance is 11.22%. This means that without other factors, medical insurance exerts an 11.2% influence on-farm labor participation rate. The t-value of average treatment on treated passes significance tests. Estimates of the average treatment effect of household agricultural work participation rate show that health insurance can increase the household agricultural work participation rate by 12.79%. The net effect of the non-farm labor participation rate brought by health insurance participation is −0.0651, and the t-value of the average treatment effect passes the significance test. The average treatment effect estimate of the labor participation rate of the self-employed activities shows that health insurance can reduce the labor participation rate of the self-employed activities of farmers by 6.49%. The PSM estimation result is similar to the Probit estimation result. This result verifies that after controlling for the influences of other factors, medical insurance systems encourage farmers to reduce their participation of self-employed labor and non-agricultural labor, and transfer to rural areas to engage in household agricultural work labor.

### 4.4. Model Estimation of Labor Participation in Different Health Insurance Projects

Different health insurance systems have different effects on the labor participation rate of middle-aged and elderly peasant households. At present, the new cooperative medical insurance covered almost all farmers is more inclined to encourage middle-aged and old farmers to improve the farm labor participation rate, especially the household agricultural labor participation rate. However, it also has a negative effect on the non-farm labor supply, especially the self-employed labor participation rate. It has a positive effect on the total labor participation rate. This is probably due to the new cooperative medical insurance only permits local reimbursement, which encourages farmers to leave the urban region and return to the farming area. At the same time, the new cooperative medical insurance raises human capital and increases farmers’ household agricultural labor supply. From Table 10, household agricultural work labor participation rate in the new cooperative medical insurance increased by 13.74%, the farm labor participation rate increased by 13.00%, and the total labor participation rate increased by 3.39%. The non-farm labor participation rate decreased by 6.49%, and the self-employed labor participation rate decreased by 5.29%. The new cooperative medical insurance has a more potent pull-back effect for middle-aged and elderly farmers. It encourages middle-aged and elderly farmers to leave the non-farm sectors, and transfer to agricultural and rural areas to engage in household agricultural labor. The urban employee medical insurance encourages farmers to reduce self-employed labor supply and increase employed work. It is probably because the urban employee medical insurance pays higher proportions in claims and holds higher personal account values. It encourages farmers to reduce labor supply in the self-employed labor supply and increase employed labor supply. Urban resident medical insurance incentivizes migrant farmers to abandon self-employed labor supply and withdraw from the labor market. It is probably due to the target of urban resident medical insurance being unemployed individuals. With subsidies, this scheme shows a strong income effect on migrant farmers that encourages them to reduce self-employed work and exit the labor market. The urban employee medical insurance encourages farmers to reduce self-employed labor supply and increase employed work as a result of a high level of reimbursement paid in claims and high personal account values. The government medical insurance has significant adverse effects on the labor participation rate and encourages them to withdraw directly from the labor market. It is probably because government employees do not consider coverage provided by other part-time jobs due to the superiority of the government medical insurance scheme. The government raises funds for this scheme, and individuals do not have to pay insurance premiums. Therefore, government medical insurance shows a strong income effect that encourages them to withdraw from the labor market. Supplementary health insurance tends to reduce the farm employed labor participation rate and non-farm labor participation rate, especially the self-employed sector, and also tends to increase the farm labor supply, especially household agricultural work. Similar to new cooperative medical insurance which only permits local claims, supplementary health insurance also prohibits cross-regional claims. It incentivizes farmers to leave the urban region and return to the rural farming area. The consequences are a reduction of self-employed labor supply, elevated human capital, and encourages farmers to increase their household agricultural labor supply.

Table 10 shows that the new cooperative medical insurance and supplement health insurance significantly increase agricultural labor supply, whereas urban employee medical insurance, urban resident medical insurance, and government medical insurance’s incentive effect are not prominent. Therefore, the overall combined effect of all five medical insurance schemes is health insurance significantly increases farmers’ household agricultural labor supply. The new cooperative medical insurance and Supplementary health insurance improve farmers’ health and raise human capital by increasing medical expenditure. This in turn raises working capabilities and labor returns that incentivize farmers to increase labor participation and labor supply. The new cooperative medical insurance and supplement health insurance exerted a substitution effect similar to the wage rate by increasing the marginal return of labor. This effect will have a positive impact on the labor supply. As a result, overall health insurance poses a much stronger substitution effect on labor supply that encourages farmers to increase labor supply. Furthermore, the new cooperative medical insurance, urban employee medical insurance, urban resident insurance, and supplementary health insurance all incentivize farmers to reduce self-employed labor supply, whereas the government medical insurance show no significant influence on farmers’ self-employed labor supply. Thus, the combined effect of health insurance is a reduction in labor supply. The reasons for new cooperative medical insurance, urban employee medical insurance, urban resident insurance, and supplement health insurance all incentivize farmers to reduce self-employed labor supply may be related to the inherent characteristics of each scheme. The new cooperative medical insurance and the supplement health insurance only permit local claims and prohibit cross-regional claims. This rule means that farmers are unable to enjoy the benefits of medical insurances in the urban region. It will encourage them to leave urban regions and return to rural farming areas. Urban resident medical insurance targets unemployed individuals that encourage farmers to engage in self-employed labor supply to withdraw from the labor market. Urban employee medical insurance targets employed workers. Its high value encourages farmers to switch from self-employed labor supply to employed works. These reasons ultimately lead to the combined effect of health insurances to reduce farmers’ labor supply in self-employed labor supply.

### 4.5. The Influence of Health Insurance on the Labor Supply Time of the Middle and Old Aged Farmers

The effect of health insurance coverage on the labor supply time behavior is not apparent. In terms of subdivision, health insurance coverage significantly increases the farm labor supply time and significantly reduces the non-farm labor supply time. Further observation of different types of labor supply models shows that health insurance coverage significantly increases the household agricultural labor supply time and reduces the self-employed labor supply time, which has no significant impact on the farm employed labor supply time and the employed labor supply time.

From the estimated results (Table 11), the significance of the variables of health insurance in the household agricultural work supply model (LSf2) and the labor supply of farm model (LSf) is tested at 1%. The estimated marginal effects are all positive, indicating that participating in health insurance would tend to increase the farm labor supply time and household agricultural labor supply time. From the perspective of the Tobit marginal effect, participating in health insurance will increase the household agricultural work supply time by 197.71 h per year, and increase the farm labor supply time by 200.70 h per year. In the self-employed labor supply model (LSn2) and the non-farm (LSn) labor supply time, the coverage variables of health insurance are tested on the significance of 1%. The estimated marginal effects are all adverse. It shows that under the control of other factors, participating in health insurance tends to reduce the self-employed labor supply time and the labor supply time of non-farm. From the perspective of the Tobit marginal effect, participating in health insurance reduces the self-employed labor supply time by 198.9 h per year and non-farm labor supply time by 143.33 h per year. In the total labor supply model (TLS), farm employed labor supply model (LSf1), and employed labor supply model (LSn1), the variables of health insurance coverage do not pass the significant test, indicating that participating in health insurance do not affect the total labor supply time behavior, farm employed labor supply time behavior, and employed labor supply time behavior.

Similarly, considering the potential endogeneity issue with insurance coverage and labor supply, IVtobit method with whether or not the village carried out new cooperative medical insurance as an instrument is used for estimation (Table 12). The *p*-value of Wald test is less than 0.05 showing that the endogeneity problem does exist. However, the direction of the results of IVtobit and Tobit method is the same. The differences are merely marginal effects. Therefore, although the endogeneity problem does exist with insurance coverage and labor supply, it does not hinder the validity of the results.

Similar to the endogenous problem between health insurance coverage and labor participation in decision-making behavior, the endogenous problem of health insurance coverage and labor supply time behavior and its impact on the conclusion is still illustrated by using PSM. Using the PSM method, we can see the results (Table 13). Under the condition of paired matching between treatment and control group, the net effect of farm labor supply time brought by medical insurance is 154.86 h per year, and the net effect of household agricultural labor supply time is 183.17 h per year, the t-values of average treatment effect pass the significance test. Participation in medical insurance reduced the non-farm labor supply of farmers by 157.74 h per year, and the labor supply of self-employed labor of farmers by 153.20 h per year. The t-value of the average treatment effect passed the significance test. Although there is an endogenous problem between health insurance coverage and labor supply time behavior, the direction of effects is essentially the same for PSM and Tobit models. The difference lies in the different marginal effects, so it does not affect the correctness of its conclusion.

Besides, the impact of different types of health insurance on farmers’ labor supply time also has significant differences. The estimated results of the model are consistent with those of the previous labor participation model, and the specific results are not reported.

## 5. Conclusions

This paper examines the effect of China’s health insurance on labor supply using data from the CHARLS. The results reveal that health insurance tends to encourage farmers to improve farm labor, especially household agricultural labor supply. However, a negative effect is noted on the non-farm labor supply, especially on the self-employed economy. From this perspective, health insurance encourages farmers to leave urban areas and stay in rural areas to engage in household agricultural labor. Moreover, various types of health insurance have different effects on labor supply. The new cooperative medical insurance has a more substantial pull-back effect, which encourages farmers to leave the non-farm sector and transfer to the agricultural and rural areas to engage in household agricultural work. The urban employee medical insurance encourages farmers to reduce self-employed labor supply and increase employed work. The urban resident medical insurance encourages farmers to reduce self-employed labor supply or even quit the labor market. The government medical insurance encourages farmers to quit directly from the labor market. Similar to new cooperative medical insurance, supplemental health insurance tends to encourage the farmers to leave the non-farm sector and transfer to the rural areas to engage in household agricultural work. However, it tends to improve the farm labor supply, especially the self-employed labor supply. A previous focus on framework design and system exploration in China’s health insurance system is identified. The research and policy departments prioritize the institutional design and internal operation of the system without considering its profound impact on the entire economic and social system. The health insurance system has a noticeable effect on labor supply, which has critical policy implications for the future reform of the health insurance system, urban and rural labor market organization, and long-term economic growth.

First, the government should be cognizant of the effect on the economic development of accelerating health insurance reform. Currently, health insurance is changing the structure of the labor market, encouraging farmers to leave urban and return to rural areas to engage in their household agricultural work, which may further aggravate the shortage of migrant workers. So, China’s economic development might suffer. The supply effect of different health insurance projects also differs, which might be the result of China’s health insurance system fragmentation and system immaturity. It poses challenges to the convergence and integration of the urban and rural resident medical insurance in the process of urbanization as well as the convergence and unification between different groups.

Second, the health care system significantly affects the labor market. As a new cooperative medical insurance with the broadest coverage, the fastest development, and the most profound effect on farmers, it has resulted in a more substantial pull-back effect, encouraging rural farmers to leave the urban non-farm sector and transfer to agricultural and rural areas to engage in household agricultural work. This movement aggravates the labor shortage and indicates that the new cooperative medical insurance accelerates the transformation of the labor market. In addition to improving the health insurance system itself, the policy-making department should also thoroughly consider the impact of the health insurance system on the labor market.

Third, the government should develop a health insurance system that is compatible with the labor market and the country’s economic development. China’s medical insurance is changing the labor market, thus affecting economic development. We must establish a reasonable medical insurance system to maximize the overall welfare and have appropriate protection. The sustainable development of Chinese people’s life and welfare must consider the sustainable development of Chinese people’s welfare and welfare.

## Figures and Tables

**Table 1 ijerph-17-06689-t001:** Variable definition and descriptive statistics.

Variable	Variable Definitions	Assignment	Mean	SD	Min	Max
Explained variable						
TLP	total labor participation	1 Yes, 0 No	0.82	0.38	0.0	1.0
LPf	farm labor participation	1 Yes, 0 No	0.68	0.46	0.0	1.0
LPf1	farm employed labor participation	1 Yes, 0 No	0.06	0.23	0.0	1.0
LPf2	household agricultural labor participation	1 Yes, 0 No	0.63	0.48	0.0	1.0
LPn	non-farm labor participation	1 Yes, 0 No	0.28	0.45	0.0	1.0
LPn1	employed labor participation	1 Yes, 0 No	0.18	0.39	0.0	1.0
LPn2	self-employed participation	1 Yes, 0 No	0.11	0.31	0.0	1.0
TLS	total labor supply time	hour/year	1315.8	1334.0	0.0	6471.3
LSf	farm labor supply time	hours per year	762.4	1024.2	0.0	6111.8
LSf1	farm employed labor supply time	hours per year	26.5	175.8	0.0	3595.2
LSf2	household agricultural labor supply time	hours per year	735.9	1005.4	0.0	6111.8
LSn	non-farm labor supply time	hours per year	553.4	1138.7	0	6471.3
LSn1	employed labor supply time	hours per year	336.0	880.3	0.0	6471.3
LSn2	self-employed supply time	hours per year	217.3	808.9	0.0	6471.3
Explanatory variables						
Health insurance variables						
hicov	Health insurance coverage	1 Yes, 0 No	0.95	0.19	0.0	1.0
UEMI	Urban employee medical insurance	1 Yes, 0 No	0.02	0.13	0.0	1.0
URMI	Urban resident medical insurance	1 Yes, 0 No	0.01	0.09	0.0	1.0
NCMI	New cooperative medical insurance	1 Yes, 0 No	0.91	0.28	0.0	1.0
GMI	Government medical insurance	1 Yes, 0 No	0.003	0.05	0.0	1.0
SHI	Supplement health insurance	1 Yes, 0 No	0.07	0.25	0.0	1.0
Personal characteristics variables						
sex	gender	1 male, 0 female	0.47	0.50	0.0	1.0
age	age	year old	55.2	5.85	45.0	65.0
agesq	age squared	age squared	3079.7	645.9	2025	4225
edu	education level	year	4.36	4.3	0.0	16.0
married	marital status	1 married, 0 unmarried	0.93	0.25	0.0	1.0
party	party	1 Yes, 0 No	0.06	0.25	0.0	1.0
adl/iadl	state of health	1 adl/iadl, 0 No	0.19	0.39	0.0	1.0
Family variables						
household	family size	people	5.62	1.89	1.0	17.0
dependency	family dependency ratio	%	0.16	0.31	0.0	3.0
income	total income	ten thousand RMB per year	0.70	2.61	−9.6	134.5
incomef	farm income	ten thousand RMB per year	0.23	2.35	−10.5	134.5
Incomef1	farm employed income	ten thousand RMB per year	0.02	0.10	0	1.5
Incomef2	household agricultural income	ten thousand RMB per year	0.22	2.34	−10.5	134
incomen	non-farm income	ten thousand RMB per year	0.47	1.17	−10	15.0
Incomen1	employed income	ten thousand RMB per year	0.34	0.95	0	10
Incomen2	self-employed income	ten thousand RMB per year	0.13	0.68	−10	10
land	cultivated area	Mu	15.20	2.02	0	13,000
house	own home value	ten thousand RMB	0.22	5.73	0	300
value	consumer durables value	ten thousand RMB	5.28	10.8	0	190.8
money	total bank deposits and cash	ten thousand RMB	−0.46	6.52	−289.9	75.2
re_support	get financial support outside the family	ten thousand RMB per year	0.46	1.83	0	100.0
e_support	provide family financial assistance	ten thousand RMB per year	0.45	1.74	0	50.25
Regional variables						
D1	east	1 Yes, 0 No	0.35	0.47	0.0	1.0
D2	middle	1 Yes, 0 No	0.36	0.48	0.0	1.0
D3	west	1 Yes, 0 No	0.28	0.45	0.0	1.0

**Table 2 ijerph-17-06689-t002:** Health insurance coverage and labor force participation rate (%) in 2013.

		TLP	LPf	LPf1	LPf2	LPn	LPn1	LPn2
Peasant household sample	covered	82.19	68.82	5.91	63.83	28.44	18.39	10.37
	uncovered	77.27	55.56	6.17	50.12	35.31	19.01	16.55

**Table 3 ijerph-17-06689-t003:** Health insurance coverage and labor supply time (hourly/year) in 2013.

		TLS	LSf	LSf1	LSf2	LSn	LSn1	LSn2
Peasant household sample	covered	1318.5	772.2	26.4	745.8	546.2	334.7	211.4
	uncovered	1258.3	536.0	27.8	508.1	722.3	363.3	358.9

**Table 4 ijerph-17-06689-t004:** Different types of health insurance coverage and labor participation rate (%).

	TLP	LPf	LPf1	LPf2	LPn	LPn1	LPn2
New cooperative medical insurance	82.46	69.78	6.03	64.68	27.80	17.74	10.35
Urban employee medical insurance	82.94	40.00	1.18	37.65	65.88	58.82	7.65
Urban resident medical insurance	72.73	44.44	3.03	42.42	37.37	24.24	13.13
Government medical insurance	82.61	43.48	0.00	39.13	60.87	47.83	13.04
Supplement health insurance	82.48	70.07	7.15	65.99	24.96	17.66	7.30
uncovered	77.27	55.56	6.17	50.12	35.31	19.01	16.55

**Table 5 ijerph-17-06689-t005:** Different types of health insurance coverage and labor supply time (hours/year).

	TLS	LSf	LSf1	LSf2	LSn	LSn1	LSn2
New cooperative medical insurance	1307.7	778.7	27.3	751.4	528.9	319.1	209.8
Urban employee medical insurance	1823.4	294.5	2.4	292.0	1528.8	1371.1	157.6
Urban resident medical insurance	1439.4	534.5	1.1	533.4	904.8	546.3	358.5
Government medical insurance	1115.9	142.9	0	142.9	973.0	671.5	301.4
Supplement health insurance	1322.8	848.4	26.5	821.8	474.3	339.2	135.1
uncovered	1258.3	536.0	27.8	508.1	722.3	363.3	358.9

**Table 6 ijerph-17-06689-t006:** Model estimation of labor participation in health insurance.

	(1)	(2)	(3)	(4)	(5)	(6)	(7)
	TLP	LPf	LPf1	LPf2	LPn	LPn1	LPn2
hicov	0.0285	0.1196 ***	−0.0080	0.1280 ***	−0.0617 ***	−0.0037	−0.0533 ***
	(1.563)	(5.445)	(−1.606)	(5.511)	(−3.233)	(−0.207)	(−4.124)
sex	0.0986 ***	0.0625 ***	0.0075 ***	0.0519 ***	0.1400 ***	0.0889 ***	0.0499 ***
	(12.179)	(6.325)	(2.832)	(5.024)	(16.904)	(11.958)	(7.903)
age	0.0009	0.0114	0.0052	0.0209	0.0170	0.0171	−0.0031
	(0.064)	(0.632)	(1.021)	(1.117)	(1.086)	(1.303)	(−0.267)
agesq	−0.0001	−0.0001	−0.0001	−0.0002	−0.0002 *	−0.0002 *	−0.0000
	(−0.511)	(−0.715)	(−1.133)	(−1.182)	(−1.739)	(−1.758)	(−0.128)
edu	−0.0025 **	−0.0061 ***	−0.0002	−0.0041 ***	0.0038 ***	0.0007	0.0034 ***
	(−2.532)	(−5.164)	(−0.634)	(−3.318)	(3.749)	(0.789)	(4.473)
married	0.0707 ***	0.0912 ***	−0.0030	0.0919 ***	−0.0217	−0.0163	−0.0022
	(5.164)	(5.070)	(−0.637)	(4.859)	(−1.309)	(−1.265)	(−0.164)
party	−0.0091	0.0105	−0.0028	0.0233	0.0018	0.0247**	−0.0238*
	(−0.579)	(0.549)	(−0.553)	(1.171)	(0.115)	(1.963)	(−1.886)
adl/iadl	−0.1229 ***	−0.1075 ***	−0.0015	−0.0979 ***	−0.0946 ***	−0.0654 ***	−0.0411 ***
	(−14.232)	(−9.307)	(−0.460)	(−8.072)	(−8.146)	(−6.619)	(−4.598)
household	−0.0027	0.0046 *	0.0004	0.0046 *	−0.0016	−0.0011	0.0010
	(−1.289)	(1.741)	(0.631)	(1.689)	(−0.684)	(−0.600)	(0.588)
dependency	0.0059	−0.0323 **	−0.0012	−0.0366 **	0.0355 ***	0.0146	0.0155 *
	(0.446)	(−2.055)	(−0.284)	(−2.216)	(2.655)	(1.282)	(1.679)
income	0.0338 ***						
	(7.796)						
incomef		0.0285 **			0.0007		
		(1.986)			(0.389)		
incomen		−0.0554 ***			0.1240 ***		
		(−12.354)			(19.051)		
incomef1			—	0.2639 ***		0.0037	−0.0326
			—	(3.843)		(0.117)	(−0.934)
incomef2			0.0005	0.0225 **		−0.0037	0.0016
			(1.068)	(2.269)		(−1.005)	(1.641)
incomen1			−0.0025 *	−0.0539 ***		0.1111 ***	−0.0175 ***
			(−1.720)	(−9.872)		(18.502)	(−4.607)
incomen2			−0.0020	−0.0602 ***		−0.0428 ***	0.0609 ***
			(−0.860)	(−6.631)		(−3.890)	(11.380)
land	−0.0016	0.0074	−0.0016 **	0.0007	−0.0212 ***	−0.0097	−0.0169 *
	(−0.860)	(0.965)	(−2.227)	(0.278)	(−2.748)	(−1.632)	(−1.883)
house	−0.0001	−0.0062 **	−0.0000	−0.0057 **	0.0015 **	−0.0011	0.0012 ***
	(−0.298)	(−2.096)	(−0.523)	(−1.965)	(2.030)	(−0.950)	(2.623)
value	−0.0020 ***	−0.0041 ***	−0.0003	−0.0042 ***	0.0004	0.0002	0.0007 **
	(−5.403)	(−7.011)	(−1.514)	(−6.420)	(0.907)	(0.493)	(2.370)
money	0.0001	0.0005	−0.0002 **	0.0001	0.0006	0.0006	0.0002
	(0.101)	(0.626)	(−2.187)	(0.158)	(0.901)	(0.763)	(0.433)
re_suppor	−0.0052 *	−0.0073 **	0.0001	−0.0093 **	−0.0006	−0.0027	0.0012
	(−1.741)	(−2.254)	(0.278)	(−2.432)	(−0.226)	(−1.336)	(0.844)
e_support	−0.0022	−0.0006	−0.0007	0.0006	−0.0028	−0.0025	0.0008
	(−1.104)	(−0.229)	(−0.637)	(0.225)	(−1.106)	(−1.378)	(0.564)
D2	0.0010	0.0356 ***	0.0032	0.0437 ***	−0.0444 ***	−0.0391 ***	−0.0006
	(0.113)	(3.357)	(1.160)	(3.944)	(−4.759)	(−5.021)	(−0.085)
D3	0.0706 ***	0.1245 ***	0.0011	0.1302 ***	−0.0817 ***	−0.0679 ***	−0.0132 *
	(7.249)	(10.634)	(0.370)	(10.769)	(−7.918)	(−7.712)	(−1.699)
*Pseudo R* ^2^	0.0921	0.0627	0.0227	0.0547	0.2315	0.2646	0.1154
Observations	9875	9875	9412	9875	9875	9875	9875

Notes: This table presents the probit regression results. The T statistic is in parentheses under the coefficient. *, ** and *** mean statistical significance at 10%, 5%, and 1%, respectively.

**Table 7 ijerph-17-06689-t007:** Estimation of labor participation model in medical insurance (Ivprobit).

	(1)	(2)	(3)	(4)	(5)	(6)	(7)
	TLP	LPf	LPf1	LPf2	LPn	LPn1	LPn2
hicov	14.9231	27.6438 ***	8.0043	24.2918 ***	−9.0682 **	−5.7299	−8.0764 **
	(0.591)	(2.751)	(0.847)	(2.715)	(−2.229)	(−1.599)	(−2.047)
sex	0.5202 ***	0.3590 ***	0.2663 ***	0.3023 **	0.4814 ***	0.4167 ***	0.2609 ***
	(6.377)	(2.668)	(2.725)	(2.499)	(8.729)	(8.434)	(4.593)
age	0.0136	0.0661	0.1907	0.0896	0.0512	0.0775	−0.0308
	(0.103)	(0.300)	(1.171)	(0.459)	(0.557)	(0.947)	(−0.325)
agesq	−0.0006	−0.0011	−0.0020	−0.0012	−0.0007	−0.0009	0.0002
	(−0.520)	(−0.537)	(−1.323)	(−0.693)	(−0.807)	(−1.189)	(0.174)
edu	−0.0359 ***	−0.0665 ***	−0.0219	−0.0535 ***	0.0299 ***	0.0133	0.0344 ***
	(−2.735)	(−2.905)	(−1.048)	(−2.640)	(3.209)	(1.629)	(3.727)
married	−0.1878	−0.6262	−0.3711	−0.5284	0.2126	0.1046	0.2389
	(−0.808)	(−1.555)	(−1.026)	(−1.477)	(1.286)	(0.713)	(1.449)
party	−0.1615	−0.1869	−0.1512	−0.1203	0.0751	0.1659 **	−0.0852
	(−1.111)	(−0.771)	(−0.813)	(−0.562)	(0.763)	(1.998)	(−0.830)
adl/iadl	−0.3519 ***	−0.0002	0.0334	0.0085	−0.4707 ***	−0.4030 ***	−0.3511 ***
	(−3.251)	(−0.001)	(0.244)	(0.051)	(−5.882)	(−5.478)	(−4.262)
household	−0.0136	0.0090	0.0106	0.0087	−0.0042	−0.0044	0.0084
	(−0.738)	(0.290)	(0.474)	(0.316)	(−0.324)	(−0.377)	(0.628)
dependency	0.0228	−0.0784	−0.0362	−0.0860	0.1203	0.0654	0.0802
	(0.194)	(−0.400)	(−0.253)	(−0.496)	(1.490)	(0.930)	(0.981)
income	0.1386 ***						
	(7.453)						
incomef		0.0606 **			0.0111		
		(2.133)			(1.104)		
incomen		−0.1240 **			0.4609 ***		
		(−2.329)			(19.750)		
incomef1			0.0051	0.7576		0.0157	−0.2038
			(0.003)	(1.523)		(0.083)	(−0.826)
incomef2			−0.0002	0.0408		−0.0137	0.0172*
			(−0.005)	(1.619)		(−0.908)	(1.770)
incomen1			−0.0646	−0.1338 **		0.5633 ***	−0.1138 ***
			(−1.275)	(−2.462)		(26.971)	(−4.419)
incomen2			−0.0296	−0.0910		−0.2399 ***	0.3501 ***
			(−0.336)	(−1.108)		(−6.070)	(9.725)
land	−0.0148	0.0084	−0.0506	−0.0079	−0.0768 ***	−0.0466 *	−0.0995 **
	(−0.819)	(0.258)	(−0.765)	(−0.316)	(−3.081)	(−1.758)	(−2.283)
house	−0.0016	−0.0190	−0.0017	−0.0164	0.0061	−0.0054	0.0079 **
	(−0.284)	(−1.468)	(−0.197)	(−1.383)	(1.634)	(−0.580)	(2.244)
value	−0.0132 ***	−0.0233 ***	−0.0132 *	−0.0220 ***	0.0050 *	0.0034	0.0071 ***
	(−3.626)	(−3.514)	(−1.773)	(−3.595)	(1.833)	(1.391)	(2.654)
money	0.0013	0.0033	−0.0050	0.0023	0.0017	0.0029	0.0010
	(0.269)	(0.383)	(−1.002)	(0.297)	(0.475)	(0.887)	(0.291)
re_support	−0.0402 **	−0.0562 *	−0.0054	−0.0567 *	0.0088	−0.0070	0.0171
	(−2.110)	(−1.668)	(−0.218)	(−1.877)	(0.686)	(−0.505)	(1.366)
e_support	−0.0352	−0.0492	−0.0362	−0.0408	0.0048	−0.0028	0.0189
	(−1.645)	(−1.340)	(−0.926)	(−1.249)	(0.315)	(−0.195)	(1.299)
D2	−0.2801 **	−0.4034 *	−0.0622	−0.3291	−0.0013	−0.0917	0.1464
	(−2.066)	(−1.754)	(−0.303)	(−1.602)	(−0.014)	(−1.112)	(1.573)
D3	0.1506	0.1145	−0.0528	0.1345	−0.2283 ***	−0.2930 ***	−0.0058
	(1.455)	(0.664)	(−0.371)	(0.874)	(−3.215)	(−4.616)	(−0.080)
_cons	−11.8203 *	−25.3425 **	−13.8319	−23.2767 **	6.6765	2.5704	7.0616
	(−1.890)	(−2.344)	(−1.360)	(−2.419)	(1.512)	(0.657)	(1.614)
Wald test	71.90	151.82	1.08	128.51	9.54	4.36	6.23
	[0.000]	[0.000]	[0.298]	[0.000]	[0.002]	[0.036]	[0.012]
Observations	9875	9875	9412	9875	9875	9875	9875

Notes: The IVprobit estimation results report the regression coefficient, the values in brackets are t values, and the values in square brackets represent the chi-square test *p*-value of Wald’s exogenous experience. *, ** and *** mean statistical significance at 10%, 5%, and 1%, respectively.

**Table 8 ijerph-17-06689-t008:** Estimation of Logit for farmers’ participation in health insurance decision making.

hicov	Coef.	Std. Err.	z	P > z
sex	−0.15918	0.1113527	−1.43	0.153
age	−0.0611322	0.2069792	−0.3	0.768
agesq	0.0009926	0.0018843	0.53	0.598
edu	0.0434378	0.0140018	3.1	0.002
married	0.6350918	0.1723208	3.69	0
party	0.2859729	0.2640138	1.08	0.279
adl/iadl	−0.2688632	0.1272882	−2.11	0.035
household	0.0087767	0.0299166	0.29	0.769
dependency	−0.0000114	0.1768849	0	1
income	0.0222835	0.0361534	0.62	0.538
land	0.1063997	0.139598	0.76	0.446
house	0.0013049	0.0116353	0.11	0.911
value	0.0104966	0.0071861	1.46	0.144
money	−0.0030154	0.0102559	−0.29	0.769
re_support	0.1908918	0.0893492	2.14	0.033
e_support	0.1467006	0.0850253	1.73	0.084
D2	0.4888073	0.1288465	3.79	0
D3	0.2454907	0.1269231	1.93	0.053
_cons	2.42639	5.638717	0.43	0.667
Observations	9901			

**Table 9 ijerph-17-06689-t009:** ATT of health insurance to the labor participation rate.

Types of Labor Participation	Untreated	Treated	ATT	S.E.	T-Stat
Total labor participation rate (TLP)	393	9405	0.0226	0.0252	0.90
Labor participation rate of farm (LPf)	393	9387	0.1122	0.0304	3.69
Farm employed labor participation rate (LPf1)	393	9383	−0.0057	0.0171	−0.34
Household agricultural work labor participation rate (LPf2)	393	9383	0.1279	0.0308	4.15
Labor participation rate of non-farm (LPn)	393	9387	−0.0651	0.0294	−2.21
Employed labor participation rate (LPn1)	393	9383	0.0030	0.0246	0.12
Self-employed labor participation rate(LPn2)	393	9383	−0.0649	0.0229	−2.83

**Table 10 ijerph-17-06689-t010:** Model estimation of labor participation in different types of health insurance.

	(1)	(2)	(3)	(4)	(5)	(6)	(7)
	TLP	LPf	LPf1	LPf2	LPn	LPn1	LPn2
New cooperative medical insurance	0.0339 *	0.1300 ***	−0.0079	0.1374 ***	−0.0649 ***	−0.0072	−0.0529 ***
	(1.897)	(6.030)	(−1.605)	(5.996)	(−3.424)	(−0.413)	(−4.096)
*Pseudo R* ^2^	0.0900	0.0595	0.0234	0.0515	0.2240	0.2535	0.1170
Urban resident medical insurance	−0.0923 **	−0.0352	−0.0049	0.0189	−0.0738	0.0117	−0.0772 *
	(−2.108)	(−0.619)	(−0.300)	(0.330)	(−1.614)	(0.329)	(−1.766)
*Pseudo R* ^2^	0.1388	0.1135	0.1005	0.1433	0.3164	0.3307	0.2795
Urban employee medical insurance	−0.0459	−0.0433		−0.0203	0.0373	0.1513 ***	−0.1357 ***
	(−1.107)	(−0.849)		(−0.394)	(0.890)	(4.419)	(−3.676)
*Pseudo R* ^2^	0.1534	0.1007	0.1489	0.1145	0.3621	0.4014	0.2304
Government medical insurance	−0.0302 *	0.0054		0.0225	0.0427	0.0466	0.0042
	(−1.759)	(0.042)		(0.185)	(0.357)	(0.549)	(0.064)
*Pseudo R* ^2^	0.1257	0.1269	0.1489	0.1622	0.3228	0.3352	0.3425
Supplementary health insurance	0.0023	0.0994 ***	−0.0171 **	0.1175 ***	−0.0839 ***	−0.0103	−0.0725 ***
	(0.098)	(3.520)	(−2.342)	(4.079)	(−3.489)	(−0.498)	(−4.080)
*Pseudo R* ^2^	0.1124	0.1162	0.1602	0.1214	0.2751	0.3484	0.1935

Notes: This table presents the probit regression results. The T statistic is in parentheses under the coefficient. *, ** and *** mean statistical significance at 10%, 5%, and 1%, respectively.

**Table 11 ijerph-17-06689-t011:** Estimation of labor supply model for health insurance.

	(1)	(2)	(3)	(4)	(5)	(6)	(7)
	TLS	LSf	LSf1	LSf2	LSn	LSn1	LSn2
hicov	44.97	200.70 ***	−0.24	197.71 ***	−143.33 ***	−13.49	−198.90 ***
	(0.966)	(5.621)	(−0.111)	(5.652)	(−3.304)	(−0.328)	(−3.760)
sex	282.82 ***	86.81 ***	−2.41 ***	85.75 ***	278.78 ***	194.56***	183.68 ***
	(14.625)	(6.053)	(−2.636)	(6.076)	(14.225)	(10.698)	(7.220)
age	−7.39	22.25	−0.17	19.73	26.62	42.18	−25.18
	(−0.206)	(0.844)	(−0.103)	(0.764)	(0.734)	(1.269)	(−0.535)
agesq	−0.11	−0.21	0.00	−0.19	−0.46	−0.52 *	0.06
	(−0.326)	(−0.883)	(0.097)	(−0.792)	(−1.391)	(−1.702)	(0.136)
edu	0.47	−7.55 ***	−0.06	−6.80 ***	10.55 ***	1.95	14.64 ***
	(0.200)	(−4.320)	(−0.560)	(−3.977)	(4.571)	(0.925)	(4.857)
married	153.16 ***	129.62 ***	1.23	127.70 ***	−24.17	−40.53	11.06
	(4.070)	(4.701)	(0.711)	(4.728)	(−0.614)	(−1.138)	(0.209)
party	−12.04	17.51	1.35	19.12	8.73	50.39*	−66.88
	(−0.324)	(0.640)	(0.771)	(0.713)	(0.250)	(1.652)	(−1.386)
adl/iadl	−237.68 ***	−91.80 ***	0.74	−86.27 ***	−246.95 ***	−166.04 ***	−193.33 ***
	(−9.895)	(−5.300)	(0.675)	(−5.087)	(−8.871)	(−6.377)	(−5.257)
household	−6.87	3.24	0.17	1.89	−3.98	−2.25	3.48
	(−1.354)	(0.869)	(0.722)	(0.519)	(−0.767)	(−0.467)	(0.527)
dependency	96.30 ***	16.73	0.99	1.69	63.87**	41.20	42.58
	(3.015)	(0.706)	(0.660)	(0.073)	(2.085)	(1.479)	(1.079)
income	38.33 ***						
	(11.264)						
incomef		18.59 ***			−0.01		
		(7.064)			(−0.004)		
incomen		−116.8 ***			200.73 ***		
		(−16.447)			(29.147)		
incomef1			575.75 ***	166.44 ***		−41.46	−140.77
			(115.691)	(2.640)		(−0.541)	(−1.119)
incomef2			−0.64 ***	16.33 ***		−10.21	3.97
			(−3.588)	(6.327)		(−1.477)	(1.217)
incomen1			−0.11	−122.36 ***		238.63 ***	−55.37 ***
			(−0.247)	(−15.116)		(33.061)	(−4.634)
incomen2			0.42	−98.26 ***		−85.41 ***	199.60 ***
			(0.650)	(−7.833)		(−5.421)	(16.002)
land	1.53	9.32 ***	0.07	9.79 ***	−48.00 ***	−21.83 *	−62.40 **
	(0.357)	(3.069)	(0.328)	(3.298)	(−3.386)	(−1.757)	(−2.281)
house	−0.81	−6.18 *	−0.01	−5.82 *	2.88 **	−1.69	3.65 ***
	(−0.512)	(−1.811)	(−0.207)	(−1.800)	(2.380)	(−0.578)	(2.997)
value	−1.91 **	−7.55 ***	−0.08 **	−7.24 ***	0.53	0.22	2.31 **
	(−2.206)	(−9.324)	(−2.006)	(−9.031)	(0.682)	(0.289)	(2.488)
money	1.28	−2.21 *	−0.02	−2.11 *	2.72 **	0.32	1.45
	(0.902)	(−1.888)	(−0.261)	(−1.857)	(2.126)	(0.242)	(1.021)
re_support	−33.90 ***	−14.89 ***	−0.11	−12.80 **	−11.96*	−6.41	−13.67
	(−4.229)	(−2.618)	(−0.457)	(−2.368)	(−1.700)	(−1.114)	(−1.268)
e_support	5.53	4.05	−0.26	4.37	−0.19	−7.03	8.45
	(1.051)	(1.032)	(−1.067)	(1.141)	(−0.038)	(−1.331)	(1.512)
D2	−56.57 ***	56.74 ***	0.75	54.72 ***	−106.55 ***	−113.09 ***	6.03
	(−2.633)	(3.561)	(0.749)	(3.505)	(−5.089)	(−5.928)	(0.220)
D3	118.30 ***	249.68 ***	0.52	246.10 ***	−216.07 ***	−195.04 ***	−57.07 *
	(5.151)	(14.652)	(0.481)	(14.739)	(−9.197)	(−8.972)	(−1.880)
*Pseudo R* ^2^	0.01	0.01	0.17	0.01	0.04	0.06	0.03
Observations	9875	9875	9875	9875	9875	9875	9875

Noted: This table presents the Tobit regression results. The T statistic is in parentheses under the coefficient. *, ** and *** mean statistical significance at 10%, 5%, and 1%, respectively.

**Table 12 ijerph-17-06689-t012:** Estimation of labor supply model of medical insurance (IVtobit).

	(1)	(2)	(3)	(4)	(5)	(6)	(7)
	TLS	LSf	LSf1	LSf2	LSn	LSn1	LSn2
hicov	1190.57	26,344.69 ***	4912.51	25,539.39 ***	−2.7 × 10^4^ **	−1.4 × 10^4^	−4.0 × 10^4^ **
	(0.384)	(2.684)	(1.194)	(2.665)	(−2.451)	(−1.613)	(−2.293)
sex	536.64 ***	365.46 ***	−38.29	367.68 ***	945.86 ***	879.76 ***	879.63 ***
	(12.716)	(2.815)	(−0.811)	(2.868)	(6.269)	(7.395)	(3.494)
age	−16.26	78.57	86.32	75.19	68.24	184.36	−204.48
	(−0.240)	(0.370)	(1.191)	(0.365)	(0.272)	(0.944)	(−0.489)
agesq	−0.20	−1.15	−0.92	−1.11	−1.06	−2.12	1.45
	(−0.316)	(−0.596)	(−1.376)	(−0.588)	(−0.463)	(−1.185)	(0.380)
edu	−0.93	−63.76 ***	−10.63	−60.53 ***	87.80 ***	32.75 *	157.81 ***
	(−0.134)	(−2.866)	(−1.227)	(−2.802)	(3.483)	(1.708)	(3.868)
married	247.47 **	−547.58	−151.71	−522.48	766.16 *	237.05	1318.38 *
	(1.989)	(−1.398)	(−0.980)	(−1.368)	(1.714)	(0.686)	(1.810)
party	−35.35	−162.91	−50.88	−148.38	230.58	335.32*	−94.67
	(−0.478)	(−0.699)	(−0.598)	(−0.656)	(0.861)	(1.708)	(−0.210)
adl/iadl	−435.86 ***	96.69	−8.12	96.73	−1305.70 ***	−993.83 ***	−1660.65 ***
	(−7.477)	(0.529)	(−0.118)	(0.544)	(−5.961)	(−5.561)	(−4.524)
household	−13.47	3.55	13.89	0.21	−9.82	−8.24	31.76
	(−1.403)	(0.118)	(1.376)	(0.007)	(−0.276)	(−0.293)	(0.539)
dependency	172.99 ***	51.03	−9.76	14.62	219.68	189.87	204.56
	(2.878)	(0.270)	(−0.151)	(0.080)	(1.003)	(1.143)	(0.566)
income	71.18 ***						
	(10.740)						
incomef		18.91			25.43		
		(0.773)			(0.930)		
incomen		−233.64 ***			749.46 ***		
		(−4.498)			(13.077)		
incomef1			3220.18 ***	419.61		−212.51	−869.00
			(24.199)	(0.806)		(−0.459)	(−0.800)
incomef2			−0.22	14.78		−37.81	61.92
			(−0.034)	(0.620)		(−1.024)	(1.485)
incomen1			−53.05 **	−271.64 ***		1165.48 ***	−366.99 ***
			(−2.020)	(−4.675)		(24.867)	(−3.276)
incomen2			1.10	−152.17 *		−473.84 ***	1093.54 ***
			(0.030)	(−1.725)		(−5.090)	(7.200)
land	2.27	11.90	−49.50	13.54	−176.82 ***	−101.53	−356.94 **
	(0.276)	(0.453)	(−1.099)	(0.530)	(−2.872)	(−1.616)	(−2.103)
house	−1.57	−15.36	−14.26	−14.64	13.02	−7.36	24.67
	(−0.523)	(−1.322)	(−0.800)	(−1.306)	(1.285)	(−0.490)	(1.631)
value	−3.87 **	−28.14 ***	−14.22 ***	−27.93 ***	12.17 *	6.99	29.74 **
	(−2.003)	(−4.323)	(−3.710)	(−4.245)	(1.691)	(1.219)	(2.512)
money	2.49	−3.48	−0.23	−3.08	9.46	1.01	6.50
	(0.928)	(−0.421)	(−0.069)	(−0.383)	(1.008)	(0.131)	(0.446)
re_support	−65.22 ***	−68.01 **	−24.30	−62.06 *	−14.70	−15.30	−35.67
	(−4.189)	(−2.038)	(−1.180)	(−1.922)	(−0.350)	(−0.453)	(−0.443)
e_support	8.30	−35.55	−19.37	−34.04	44.10	−11.17	118.74 *
	(0.736)	(−1.001)	(−1.103)	(−0.981)	(1.082)	(−0.330)	(1.854)
D2	−123.46 *	−351.73	−14.50	−343.66	77.24	−301.17	777.22 *
	(−1.712)	(−1.573)	(−0.164)	(−1.567)	(0.306)	(−1.553)	(1.893)
D3	212.32 ***	340.70 **	−20.26	342.46 **	−605.62 ***	−835.50 ***	22.38
	(3.950)	(2.046)	(−0.327)	(2.099)	(−3.135)	(−5.510)	(0.070)
_cons	1059.46	−2.5 × 10^4^ **	−7586.93 *	−2.4 × 10^4^ **	21,796.20 *	5896.24	36,807.62 *
	(0.320)	(−2.365)	(−1.808)	(−2.346)	(1.830)	(0.640)	(1.911)
Wald test	5.17	85.36	1.96	84.26	19.86	3.77	13.51
	[0.023]	[0.000]	[0.161]	[0.000]	[0.000]	[0.052]	[0.000]
Observations	9875	9875	9412	9875	9875	9875	9875

Noted: The IVtobit estimation results report the regression coefficient, the values in brackets are t values, and the values in square brackets represent the chi−square test *p*-value of Wald’s exogenous experience. *, ** and *** mean statistical significance at 10%, 5%, and 1%, respectively.

**Table 13 ijerph-17-06689-t013:** ATT of health insurance to labor supply time.

Types of Labor Supply	Untreated	Treated	ATT	S.E.	T-Stat
Total labor supply time (TLS)	393	9405	3.44	86.8535	0.04
Labor supply time of farm (LSf)	393	9387	154.86	56.6351	2.73
Farm employed labor supply time (LSf1)	393	9383	−4.73	11.4872	−0.41
Household agricultural labor supply time (LSf2)	393	9383	183.17	53.8546	3.40
Labor supply time of non-farm (LSn)	393	9387	−157.74	81.0835	−1.95
Employed labor supply time (LSn1)	393	9383	−0.46	57.8360	−0.01
Self-employed labor supply time (LSn2)	393	9383	−153.20	66.2752	−2.31

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
