# Peer review of "The Effect of China’s Health Insurance on the Labor Supply of Middle-aged and Elderly Farmers"

_ijerph, 2020, doi:10.3390/ijerph17186689_

Round 1
Reviewer 1 Report
Thank you for this interesting paper. Here are some ways to strengthen it:
Abstract, Line 29: "participants in farmers to quite" appear to be typos and does not make sense
Line 80: Define what is NCMS the first time the abbreviation is used.
Line 84: "urban" and "Urban" ...use consistently one or the other
Line 97: What is "the demographic dividend"?
Line 101: What is the "middle-income trap"?
130: Capitalize
This paper will be of interest to many. It would be helpful to frame a section of the conclusions with the significance of these results on the labor supply without using the academic jargon of microeconomics. This will make it more accessible to lay as well as academic audiences.
Reviewer 2 Report
The evaluation is attached.

Round 2
Reviewer 2 Report
The study is now much clearer, self-explanatory, and concise.
- Lines 47-48 and 170: it is not true that the labor participation rates have declined in the major western economies. The key issue is that number of workers in many of the economies is projected to decline due to low fertility.
- Line 52: add word Chinese.
- Line 83: health insurance does not reduce health risks (the probability of becoming ill), but it limits the consequences by better access to healthcare and reimbursement of the costs.
- Suggestion to pages 3 and 4: reorganize the literature so that all negative results are in the same page.
- Line 121: a word is missing
- Lines 122-123: when first mentioning the substitution effect, define it (see line 260). It is simply that those who are insured have higher expected labor income if they give up leisure.
- Lines 204-205: Equation (3) is the second period constraint
- line 235: b or β?
- Line 238: basically, all the relations between the variables should be shown as equations in the model. At least the following equations are missing: L = T and the links between labor supply of the self-employed and the income from self-employment.
- There should be a link between realisation of bad health and the labor supply in the second period in the model, see lines 287-290.
- Lines 256-260: Income effect is related only to expected income, not the welfare from reducing uncertainty.
- Line 331 and and 347: it is worth mentioning that CHARLS has several waves.
- Lines 564-570: repeats the lines above.
- Lines 699-701: The study has not evaluated the welfare gains not related to labor income from health insurance. Also, the conclusion on the blind expansion is speculative and not supported by the statistical analysis. The conclusions should be directly related to the analysis done.
The main remaining problem is that some key links between the variables are missing in the theoretical model. Therefore, the model cannot be solved, or even the first order conditions characterized. After considering the additional comments above, the paper can be published without third round evaluation.
Author Response
We are very grateful to the reviewer for the revision comments. The reviewer’ s comments appear in italics, while our responses are in normal font. Our adjusted discussions are with red marks.
- Lines 47-48 and 170: it is not true that the labor participation rates have declined in the major western economies. The key issue is that number of workers in many of the economies is projected to decline due to low fertility.
Thank you very much for this suggestion. Lines 47-48 describes the situation in which labor participation rates have declined in China. To avoid confusion, we have changed the National Bureau of Statistics into China National Bureau of Statistics, see line 42. At the same time, we have included country restrictions in line 47. The number of workers in many of the economies is indeed projected to decline due to low fertility. This paper aims to investigate whether or not medical insurances will accelerate the change in the labor market. Such an aim is not in conflict with the reduction of labor population associated with the fact of a lower fertility rate.
2.Line 52: add word Chinese.
Thank you for your comment. We have corrected this issue, please see line 52.
3.Line 83: health insurance does not reduce health risks (the probability of becoming ill), but it limits the consequences by better access to healthcare and reimbursement of the costs.
According to the comment, we have removed the description of health insurance can reduce the health risks of insured individuals. The new expression became “In general, health insurance reduces expected medical expenditures and preventive savings, thereby incentivizing workers to reduce labor supply.” Please see the changes in line 72. Thank you.
4.Suggestion to pages 3 and 4: reorganize the literature so that all negative results are in the same page.
Thank you for your suggestion. We follow your suggestion in this organization comparing with putting all negative results on the same paper.
Literature reviews are organized into the following 5 categories:
(1) Health insurance encourages workers to reduce labor supply.
(2) Lowering the level of health insurance benefits may encourage workers to increase labor supply because of pressure from medical expenses.
(3) Health insurance may also stimulate an increase in labor supply.
(4) The effect of health insurance on labor supply varies among different groups.
(5) Some studies have shown that health insurance has an insignificant effect on labor supply.
5.Line 121: a word is missing
Thank you for point our mistake. We have added the missing word, please see the correction in line 106.
6.Lines 122-123: when first mentioning the substitution effect, define it (see line 260). It is simply that those who are insured have higher expected labor income if they give up leisure.
Thank you for your suggestion. According to your suggestion. We have included its definition when we first mentioned the substitution effect.
7.Lines 204-205: Equation (3) is the second period constraint
Thank you for your thorough review. We have fixed this issue.
8.line 235: b or β?
Thank you for point our typo. It should be β, from equation (1), see line 207.
9.Line 238: basically, all the relations between the variables should be shown as equations in the model. At least the following equations are missing: L = T and the links between labor supply of the self-employed and the income from self-employment.
Equation (4) is the time budget constraint where the time is allocated for farm-employment, household agricultural work, employment, self-employment time and leisure time. With the increase in agricultural labor productivity, labor will gradually shift towards sectors with higher incomes. Equation (7) is the link between the labor supply of the self-employed and the income from self-employment. Because it is very difficult to solve the model, following the methods in literature [38]. We will no longer optimize the theoretical model, focusing on explaining the mechanism of health insurance on labor decision-making from the theoretical framework, see line 209-210 for the modification.
- There should be a link between realisation of bad health and the labor supply in the second period in the model, see lines 287-290.
According to the review’ s suggestion, we have discussed the link between the realisation of bad health and the labor supply in the second period in the model, see lines 178-179.
11.Lines 256-260: Income effect is related only to expected income, not the welfare from reducing uncertainty.
Thank you for your comment. According to the comment, we have removed the descriptions of reducing uncertainty. Please see lines 219-222.
12.Line 331 and 347: it is worth mentioning that CHARLS has several waves.
Based on the national survey of CHARLS has three waves, respectively in 2011, 2013, and 2015, because the data in 2013 provide more abundant types of farmers’ part-time jobs than in 2015, this paper adopts the CHARLS of 2013, see line 289-291.
13.Lines 564-570: repeats the lines above.
Thank you. We have fixed the above problem.
14.Lines 699-701: The study has not evaluated the welfare gains not related to labor income from health insurance. Also, the conclusion on the blind expansion is speculative and not supported by the statistical analysis. The conclusions should be directly related to the analysis done.
This paper emphasizes the effect of medical insurance on labor supplies. Therefore, we have not evaluated the welfare gains not related to labor income from health insurance. Following the reviewer’ s comment, we have improved the conclusion section. Please see lines 643-645.
- The main remaining problem is that some key links between the variables are missing in the theoretical model. Therefore, the model cannot be solved, or even the first order conditions characterized.
Thank you for your suggestion. Because it is very difficult to solve the model, following the methods in literature [38]. We will no longer optimize the theoretical model, focusing on explaining the mechanism of health insurance on labor decision-making from the theoretical framework. We considered various variables when building the measurement model of the labor participation rate measurement model and labor supply time measurement model, see equation 9 and 10.